# Breeding Canola (*Brassica napus* L.) for Protein in Feed and Food

**DOI:** 10.3390/plants10102220

**Published:** 2021-10-19

**Authors:** Kenny K. Y. So, Robert W. Duncan

**Affiliations:** Department of Plant Science, University of Manitoba, Winnipeg, MB R3T 2N2, Canada; sok@myumanitoba.ca

**Keywords:** canola, rapeseed, *Brassica napus*, canola protein, plant proteins, breeding, food safety

## Abstract

Interest in canola (*Brassica napus* L.). In response to this interest, scientists have been tasked with altering and optimizing the protein production chain to ensure canola proteins are safe for consumption and economical to produce. Specifically, the role of plant breeders in developing suitable varieties with the necessary protein profiles is crucial to this interdisciplinary endeavour. In this article, we aim to provide an overarching review of the canola protein chain from the perspective of a plant breeder, spanning from the genetic regulation of seed storage proteins in the crop to advancements of novel breeding technologies and their application in improving protein quality in canola. A review on the current uses of canola meal in animal husbandry is presented to underscore potential limitations for the consumption of canola meal in mammals. General discussions on the allergenic potential of canola proteins and the regulation of novel food products are provided to highlight some of the challenges that will be encountered on the road to commercialization and general acceptance of canola protein as a dietary protein source.

## 1. Introduction

The global population is projected by the United Nations to increase beyond nine billion by the mid-21st century (Figure 1) [1] and scientists have been tasked with ensuring food security to sustain this growth [2,3,4]. Agriculture will be able to feed the global population provided improvements are made to the sustainability of current agricultural practices along with a concomitant shift in dietary preferences [4,5,6,7]. Complicating the task of food production are increasingly severe and unpredictable climatic patterns [6,8,9] as well as the continued reduction in arable land [6,8,9,10]. The concept of food security was initially defined by the United Nations at the World Food Conference of 1974 as the availability of food at reasonable prices at all times [11]. The definition of food security has since been expanded to encompass the nutritional and social aspects of food [12,13]. Dietary protein has been a focus of nutrition programs as it is often a limiting macronutrient in malnourished and food-insecure populations [14,15,16].

Dietary protein is primarily acquired through either the consumption of meat (animal protein) or legumes. Historically, meat consumption was associated with economic wealth and recent shifts in developing economies have dramatically increased meat consumption [17,18,19]. Intensive animal husbandry for meat production has been repeatedly cited as being environmentally destructive, unsustainable, and an inefficient method of converting protein feedstock into dietary protein [20,21,22,23]. The production of plant-based protein sources, such as kidney bean (*Phaseolus vulgaris* L.), requires substantially less input and generates less waste than an equivalent unit of red meat such as beef [20]. Given the large amount of feed required to produce animal protein [6], the redistribution of land from feed grain to crop production enables larger quantities of dietary protein to be produced per unit area [9,24]. Evidently, even partial replacement of animal proteins with plant-based alternatives will have considerable impact on long-term sustainability [25].

Although plant-based proteins are prominent in many cultures, European and North American diets show a proclivity for animal proteins in their diet [16,17,26]. However, a recent assessment of food supplies across 171 countries found a decrease in animal protein supply in six countries across North America, Europe, and Australia [27], suggesting a gradual replacement of animal protein with alternative proteins such as those from plants. Recent work has demonstrated that the composition of the human gut microbiome is in rapid flux with changes in diet [28], suggesting that humans are amenable to adapting to new diets with relative ease. The challenge of adopting plant-based proteins, therefore, may largely be cultural. In an effort to promote vegetable proteins as a nutritious, sustainable, and a secure alternative to animal protein, 2016 was designated as the International Year of Pulses [29]. Consumer food choices are influenced by dietary guidelines published by national food authorities, and different recommendations show varying degrees of sustainability based on the resources required for production [30]. Given their far-reaching influence, incorporating considerations for sustainability in government food recommendations may substantially affect food security for many people [31]. In Canada, a recent revision to the food guide changing a “meat and alternatives” food group to “protein foods” with an emphasis on plant-based proteins reinforced the suitability of plant proteins as a replacement for animal proteins [32]. In expanding the available sources of plant-based proteins, rapeseed/canola (*Brassica napus* L.) protein has been suggested as a possible plant-based protein source. With increasing global acreages annually (Figure 2), canola that is currently grown and processed for edible oil production generates a large quantity of protein-rich seed meal as a byproduct of the oil extraction process [33]. The ease of access to and availability of canola meal makes it a logically suitable and sustainable candidate for development into a plant-based protein for human consumption [34]. Such a proposal has been supported by both industry and public institutions worldwide, as reflected by numerous abstracts at the latest International Rapeseed Congress [35].

In addition to serving as a dietary protein supplement, the functional properties of canola protein render them useful in a diverse range of food processing applications [33,34,36,37]. In such a case, individual canola proteins with the desired functionality would need to be isolated with additional processing steps from the crude seed protein given the differences in functional properties of each protein constituent [34,38,39,40].

## 2. Transition from Rapeseed to Canola

Rapeseed is an economically important allotetraploid oilseed crop derived from *B. oleracea* L. and *B. rapa* L. [41]. Rapeseed was introduced into Canada in 1943 initially as a source of vegetable oil for industrial applications [42,43]. Despite the high oil content of rapeseed and its adaptability to be grown in Canada, the high percentage of erucic acid in rapeseed oil (approximately 40%) and high levels of glucosinolates (80 μmol/g seed) rendered the oil unfit for the human diet [44,45]. Erucic acid consumption was implicated in the cause of multiple pathologies in various animal models and thus its elimination was a prerequisite for the adoption of rapeseed oil for human consumption [46,47]. Glucosinolates are generally deemed innocuous when intact; however, their degradation products have been implicated as the causal agent of various health problems and are pungent, rendering both the oil and meal unpalatable [48]. Specifically, studies in rats repeatedly showed glucosinolates to adversely affect thyroid function through antagonism against iodine [49,50]. Thus, the elimination of glucosinolates was necessary before rapeseed oil could be incorporated into the human diet [51]. Canadian breeding efforts led to the development of the first low-erucic-acid and low-glucosinolate rapeseed cultivar, named Tower [52]. This and subsequent low-erucic acid, low-glucosinolate cultivars were termed “canola” in North America, and double-low or 00 rapeseed in Europe, to distinguish them from the industrial rapeseed [53]. Current commercial canola cultivars have near undetectable levels of glucosinolates (less than 20 µmol/g seed) and typically have less than 1% erucic acid, well below the maximum allowable levels of 2% erucic acid and 30 µmol glucosinolate/g seed as established by international law [53,54]. The near-elimination of erucic acid and glucosinolates deemed canola to be generally recognized as safe by the United States Food and Drug Administration (Direct Food Substances Affirmed as Generally Recognized as Safe, 21 C.F.R. §184.1555, [50 FR 3755, Jan. 28, 1985]) and is the fourth largest oilseed produced globally behind oil palm, soybean, and seed cotton (Figure 3) [55]. Domestically, canola production in Canada has grown steadily since its introduction in 1974 to approximately 20 million metric tonnes (Figure 4) and was estimated to contribute just under $30 billion to the Canadian economy [56].

Though unfit for human consumption, the potential value of erucic acid as an industrial lubricant and raw material for chemical manufacturing was realized by the USDA in the late 1960s [43,57]. At the same time, high-erucic-acid rapeseed (HEAR) cultivars with low glucosinolate levels were developed in Sweden and HEAR breeding genotypes were developed in Canada [58]. The first Canadian HEAR cultivar ‘Reston’ was registered by the University of Manitoba in 1982 (Registration number 2190).

Although canola remains primarily a crop grown for oil, its potential value as a renewable and sustainable protein source has been gaining attention. Whereas extensive breeding efforts continue to focus on improving the quality and quantity of oil in canola [59], minimal attention has been directed to improving protein quality and content due to their inverse relationship with oil content [60,61]. A canola cultivar with improved meal quality was assigned a patent to Agrigenetics Inc. and Dow AgroSciences LLC in 2016 claiming 45% crude seed protein and less than 18% fibre (US20120213909A1). To better understand the challenges of developing canola protein as a dietary protein source, we need to first consider the technical, biochemical, and genetic factors that affect proteins in canola seed.

## 3. Extracting Oil from Raw Seed

Canola/rapeseed meal is the residual portion of the seed that remains after the oil has been extracted [34]. The oil extraction process is largely divided into a pre-processing step and an extraction step [62]. Pre-processing ensures maximum oil recovery from the seed: the canola seeds are pre-heated, after which they are flattened by rollers into flakes, and subsequently cooked [63,64,65]. Heating the seeds increases the pliability of the seed to ensure they can be thoroughly flaked without shattering while the flaking and cooking steps function primarily to rupture the seed to allow oil to be released and to increase the surface area of the seed [65].

The extraction step begins with the physical removal of oil from the cooked flakes by screw press [62,65]. Remaining oil in the seed cake is then repeatedly extracted using a mixture of solvents collectively known as isohexane [65]. The meal and oil diverge in their processing after this solvent extraction; the oil is diverted for refining while the seed cake is sent to a desolventizer toaster, where a combination of steam and heat remove residual solvent from the cake [62,65,66]. The resulting oil-free meal is then typically pressed into pellets for animal feed. Alternatively, protein can be isolated from the meal. The key determinate used to differentiate canola from rapeseed is its oil quality, and their oil-free meals differ in glucosinolate content; however, such differentiation cannot be made for purified protein products from either crop. Henceforth, all protein products that are purified from meal originating from canola and rapeseed will be referred to as canola protein. The potential uses of canola protein as a food processing additive as well as non-food, non-feed applications of canola proteins have been recently reviewed [33].

## 4. Effects of Processing on Meal Quality

The quality of canola meal destined for animal feed is judged largely on its amino acid profile and digestibility [67,68]. Conversely, the quality of canola protein destined for food processing is judged on its technical functionality [37,69,70], which is conferred by the molecular structure and content of the individual proteins contained within; thus, the quality of canola protein relates to the integrity of its components.

Extraction conditions can be altered to selectively extract different protein products with different purities and technical functionalities from canola meal [40,71,72,73] and multiple patents have been assigned to different institutes to protect the intellectual property rights associated with these optimized conditions [64]. Harsh extraction processes can compromise protein quality: while seed proteins can be denatured by exposure to solvents [74], high temperatures alters the digestibility of canola meal and affects the structure of individual canola proteins [48,72,75,76,77]. Structural changes in canola proteins can abate their functional properties [78,79]. As a feed supplement, protein content is routinely determined by its nitrogen content through combustion analysis (ISO 16634-1:2008; AOAC Method 990.03) thus, denatured proteins should not affect the protein determination of the meal. However, prolonged desolventizing can negatively impact the bioavailability of essential amino acids such as lysine, due to their modification from Maillard reactions [72,80]. In addition, the digestibility of canola meal decreases with prolonged toasting, rendering it less nutritious as a feed supplement [48,75,76,77]. If the meal protein is to be used as a functional ingredient in food processing, it is crucial that the structures of the proteins are not compromised.

## 5. Nutritional Value of Canola Meal Protein

Despite the potential reduction in quality during processing, canola meal and protein both have a nutritional profile that is comparable with soy, a common plant protein source. When considering the meal as a whole, the protein content of oil-free canola meal is approximately 38% [81], compared to 44% for its soy counterpart [82] (Table 1). Canola meal tends to have high fibre, resulting in lower digestibility in animals which makes the meal less competitive compared to soy meal models [83,84]. Currently, the majority of canola meal is used for animal feed; however, isolated protein products have the potential to be used as a dietary protein source for humans [33].

Protein quality for human nutrition is most commonly measured with a Protein Digestibility Corrected Amino Acid Score (PDCAAS) [85,86,87] or, more recently, with a Digestible Indispensable Amino Acid Score [67]. The PDCAAS for a given protein is generally a ratio of its amino acid composition relative to that of a reference protein, then normalized to its digestibility; effectively, a maximum score of 1.0, indicating that one unit of the protein in question is able to supply all the essential amino acids after digestion [85,87]. When considering only the protein fraction, canola protein is comparable with its soy counterpart in its amino acid profile [68] (Table 2) and can generally satisfy human dietary requirements for essential amino acids [33,34]. The postprandial response to canola protein was empirically demonstrated to be equivalent to that of milk protein [88] and soy protein [89].

## 6. Anti-Nutrients in Canola Meal

The major drawback of canola protein nutrition is the presence of anti-nutritive compounds that negatively impact health, protein digestion, and amino acid availability [90]. The presence of glucosinolates, phytic acid, sinapine, and tannins reduce the nutritional value of canola meal [90,91].

Glucosinolates are sugar–amino acid conjugates and the content of the aliphatic class of these molecules was successfully reduced during the development of canola; however, residual quantities of phenolic glucosinolates in the meal can be broken down into off-tasting compounds that affect thyroid function [51] and show similar bioactivity to dioxins [92]. Phytic acid (phytates) is the main phosphorus-storage compound in seeds and functions as an antinutrient by binding mineral nutrients and inhibiting digestive enzymes [92]. Sinapine is an unpalatable phenolic compound that functions as an antinutrient by promoting feed-avoidance in sensitive animals [92,93]. Tannins are polyphenolic compounds which, by interacting with proteins and the gastrointestinal tract, reduce the overall digestibility of proteins and bioavailability of amino acids [90]. It should be noted that the degree of sensitivity to these compounds varies between monogastric and ruminant animals.

Progress in traditional breeding and biotechnology have led to the reduction [92,93,94] and sometimes near elimination of antinutritive compounds such as in the case of phytic acid [95]. For isolated protein products, anti-nutritive compounds are not typically considered a limitation of use as during the process of protein isolation anti-nutritive factors are excluded [96].

## 7. Seed Development in Canola

Prior to examining seed storage proteins, a general understanding of seed development is necessary. A canola seed consists of an embryo and endosperm encased in a seed coat [97]. Seed development in angiosperms is well characterized primarily based on empirical work using the model plant *Arabidopsis thaliana* (L.) Heynh. Seed development has been extensively reviewed in detail by numerous authors and will only be succinctly summarized below. Seed development in canola is broadly divided into three phases: the initial phase where rapid cell division occurs but seed growth is slow; the second phase where cells expand quickly and accumulate both storage protein and lipids; and the third phase where the seed matures and desiccates [98,99].

Embryo development begins with successful double fertilization: the fusion of sperm from the pollen with the egg forms the zygote while the endosperm results from the fusion of a second sperm with the polar nuclei [100]. The zygote then undergoes a series of coordinated divisions to establish polarity and generate tissue layers [101,102]. Embryo developmental stages are described based on their visual morphology; the stages of embryogenesis in canola have been recorded in detailed drawings from light microscopy [103,104] and scanning electron microscopy [105]. The initial globular embryo becomes heart shaped upon initiation of the cotyledon primordia; subsequently the embryo elongates into a torpedo shape; and as the cotyledons develop, they eventually bend over the embryo forming the bent cotyledon stage [103,104,106]. Although the progression from the globular stage to the bent cotyledon stage is ubiquitous across all studied canola genotypes, it is worth noting that the time of occurrence and duration of each stage shows genotypic variation [98,105,106,107,108] (Table 3).

Based on Norton and Harris [99], the first phase of seed development is characterized by expeditious growth of the hull (silique) and slow seed growth; this phase of seed development persisted for up to four weeks (28 days) after pollination when examining field-grown *B. napus* plants. When grown under controlled environments, developing seeds of the spring canola cultivar ‘Tower’ complete the first phase of seed development at approximately 23 days after pollination [98]. The second phase of seed development as described by Norton and Harris [99] occurs five to six weeks post-pollination. This phase is characterized by the initiation of embryo development and the aggregation of storage compounds within the seed. However, light microscopy [107] and scanning microscopy [105] experiments have shown embryo formation to initiate two to three days post-pollination, with globular embryos visible one week after pollination, and torpedo-shaped embryos visible 11–12 days after pollination; Norton and Harris [99] were likely referring to the expansion of the whole seed rather than the embryo. It is during this second phase of seed development that storage proteins begin to accumulate and continue to do so into the third phase of seed development at approximately 6–12 weeks after pollination [98,99].

## 8. Seed Storage Proteins of the Brassicaceae Family

Seed storage proteins (SSP) are a unique class of proteins that are specifically expressed in the developing seed and accumulated within protein storage vacuoles in the mature seed [109,110], though contemporary evidence of synthesis during germination has been reported [111]. These proteins function as a nutrient reservoir by converting nitrogen into storage-stable proteins, which are mobilized during seed germination to support early seedling growth [112,113]. Recently, SSP transcripts have been transiently detected in vegetative tissues in canola when plants were grown under different nitrogen fertility conditions [114] which corroborates the role of these proteins in nitrogen storage.

In the Brassicaceae, the seed protein pool is dominated by the globulin-type SSP cruciferin and the albumin-type SSP napin, along with minor quantities of proteins that play a diverse array of metabolic and physiological roles, respectively [110]. The globulin and albumin designation refers to the classical work of Osborne [115], who differentiated plant proteins based on their differential solubility in various solvents while the numeric values denote the sedimentation coefficient value of the protein. Late embryogenesis abundant (LEA) proteins, for example, are present in the seeds of most plants and function to mediate dehydration tolerance during seed maturation [116]. Specific to the seed protein pool of oilseed species may be substantial quantities of oleosins, proteins that enhance the stability of oil bodies in which these plants accumulate oil [117]. Extensive proteomic studies on the seeds of *B. napus* at various developmental stages have systematized the profusion of proteins present in the seed protein pool [117,118,119,120]. Furthermore, these works report remarkable plasticity in the species composition of the seed protein pool through various environmental stimuli [119,121] and physiological processes [118,120]. The spatial-temporal distribution patterns of SSP synthesis and accumulation in *B. napus* during seed development have been described in detail and exhibit genotypic variation [98,121,122,123].

### 8.1. Cruciferin of B. napus

Cruciferin is a salt-soluble globulin-type protein that accounts for approximately 60% of the total protein pool in canola seed [98]. Related 11S/12S globulin-type SSP from other crop species include cucurbitin from pumpkin (*Cucurbita maxima* Duchesne) and glycinin from soybean (*Glycine max* (L.) Merr.), with the latter being the most similar to cruciferin [110,124]. Mature cruciferin is a large hexameric protein whose subunits are each comprised of a disulfide-bridged α and β subunit. Recent work has suggested that cruciferin subunits may be reassembled into a octameric structure for storage inside protein storage vacuoles [125]. The general biosynthesis and deposition of globulin-type SSP have been extensively reviewed [126] and is summarized below.

Globulins are encoded by multigene families. Specifically, in canola, cruciferin is encoded by 9–12 genes [110]. A search on the Uniprot knowledgebase reveals five curated accessions of *B. napus* cruciferin (Table 4) (https://www.uniprot.org, accessed November 2019). Globulin-encoding genes are translated into pre-propolypeptides on the rough endoplasmic reticulum (rER) and at minimum consist of three elements (listed from N to C terminus): a signal peptide, the α subunit, and the β subunit [125,126]. The pre-propolypeptide is simultaneously translated and transported into the lumen of the rER and the signal peptide is detached, resulting in a propolypeptide [126]. As the propolypeptide is shuttled through the lumen of the rER, post-translational modifications such as glycosylation occurs followed by the formation of disulfide bridges [126]. The polypeptides are then oligomerized into entropically-favourable trimeric structures that transit to the Golgi apparatus [124,126]. After sorting into vesicles at the trans-Golgi, the trimeric structures are transported to storage protein vacuoles where the prepolypeptides are enzymatically processed by endopeptidases to yield trimers of mature globulin subunits, each consisting of an α and β chain, respectively [126,127]. Only upon production of the mature globulin subunits can mature hexameric globulins form [127]. Across different canola genotypes, variation in the temporal accumulation of both total cruciferin transcripts and proteins is observed (Table 3), thought the spatial-temporal distribution of individual cruciferin isoforms have yet to be explored. In food processing, cruciferin functions as a good gelling agent [128] and has the ability to form stronger gels than napin [129], enabling it to be used in a wide array of food products [33]. Cruciferin is also able to improve foaming stability in oil-containing mixtures [130].

### 8.2. Napin of B. napus

The second most abundant SSP in canola seed is napin, which accounts for approximately 20% of the seed protein pool [98,131]. Napin is classified as an albumin-type protein reflecting its solubility in water [115]. Low-molecular-weight proteins were initially isolated from canola by Lönnerdal and Janson [132]. Characterization of these strongly basic proteins found them to be composed of two polypeptides, approximately 90 and 30 amino acids in length, respectively, linked by disulfide bridges with a total molecular mass of 12–14 kDa [132]. Subsequent experiments demonstrated that both napin chains were generated from the cleavage of a common precursor polypeptide [133].

Seed storage albumins are encoded by multigene families [134,135]. In canola, a minimum of 10 to 16 genes were initially estimated by Southern blotting to encode napin [136,137] and no updated estimates have been published despite the availability of reference genomes. Seven manually annotated entries for *B. napus* napin are currently listed in the UniProt Knowledgebase (Table 5) (https://www.uniprot.org, accessed on November 2019).

The biosynthesis of napin mirrors that of cruciferin [126] but is initiated before the latter (Table 3) and was reviewed in detail by Mylne et al. [134]. Translation of the napin-encoding genes at the ribosomes on the rough endoplasmic reticulum (rER) generates a single pre-proalbumin polypeptide consisting of a signal peptide and the two mature napin chains with linker peptides between each element [4,5,6,7]. The signal peptide is removed during translation and the resulting proalbumin is directed into the rER lumen [134]. Inside the lumen, a pattern of eight cysteine residues conserved across similar plant storage albumins allow four disulfide bonds to be formed: two bonds are formed within the large subunit, and two intermolecular bonds are formed between the large and small subunits [134,138]. The polypeptide is then folded with the aid of chaperone proteins and can be subject to post-translational modifications [134]. Subsequently, the proalbumin is transported to the protein storage vacuole where the linker peptides are removed by proteases to form the mature protein [131,134]. The tertiary and quaternary structures of mature napin contribute to its structural stability and resistance to digestion [131,134].

Napin is water soluble, thus enabling it to be incorporated into many food products [130]. In addition, napin also has exceptional foaming capacity and good emulsifying properties [130] though these properties vary depending on the extraction process [139]. These properties enable napin to be used to partially replace more-costly milk proteins [140] and egg white [33] in food processing.

## 9. Genetic Control of Seed Storage Proteins in Canola

Given the potential value of cruciferin and napin in food processing [33,64], the development of cultivars with improved accumulation of either protein can add value to the crop. In order to facilitate breeding for improved seed storage protein in canola, knowledge of the existing genetic variation of the trait is required, in addition to an understanding of the genetic and non-genetic determinates that affect their synthesis and accumulation. To date, a single study examining genetic variation in seed storage protein was conducted in winter rapeseed [141], while similar studies have yet to be undertaken in spring rapeseed germplasm. Seed storage protein content within the seed is dependent on not only the expression of SSP-encoding genes but also on the post-translational processes that generate the quaternary structure from precursor polypeptides, and their subsequent transfer of assembled SSP into storage vacuoles [134,142,143]. The quantitative nature of SSP content suggests complex genetic regulation of the trait and indeed many genes are involved in the synthesis and processing of SSP.

Quantitative trait loci (QTL) are regions within the genome whose occurrences are associated with specific quantitative phenotypes; multiple QTL each contribute some variation to the phenotype, and multiple genes may reside within each locus [144]. The successful identification of QTL associated with important seed quality traits in canola such as oil content [145,146,147,148,149], fatty acid composition [150,151], and glucosinolate levels [146,152,153,154,155] have been reported by numerous groups. In contrast, comparatively few reports have focused on the identification of QTL associated with total seed protein in canola [145,148,149,155]. Even fewer are reports of QTL associated with protein quality traits such as SSP composition [155] and amino acid content [156].

Conserved motifs exist in the promoter regions of seed storage protein genes across diverse plant species [157] suggesting that the regulation of seed storage protein biosynthesis is governed by transcriptional mechanisms that evolved early in the evolution of the plant kingdom. The disruption of some of these conserved motifs in the canola cruciferin-encoding *cru1* gene [158] and the napin-encoding *napA* gene [159] led to reduced accumulation of the respective protein product. Multiple transcription factors are known to regulate the expression of seed storage protein genes in *Arabidopsis*: *ABSCISIC ACID INSENSITIVE 3* (*ABI3*) [160,161,162,163], *FUSCA 3* (*FUS3*) [161,164], *LEAFY COTYLEDON 1* (*LEC1)* [164,165,166], *LEC2* [161], and multiple *MYC* transcription factors [167]. A detailed summary on the transcriptional regulation of seed storage proteins was reviewed by Verdier and Thompson [168].

## 10. Non-Genetic Control of Seed Storage Proteins in Canola

Plant growth regulators are capable of modifying transcriptional activity and consequently, seed storage protein accumulation may in part be influenced by plant hormones [168]. Early works on seed storage proteins in canola implicated abscisic acid in the direct regulation of cruciferin and napin [100,169,170]. In addition to its role in inducing the accumulation of seed storage proteins, abscisic acid is also associated with abiotic stress tolerance in plants [169]. Thus, seed storage protein accumulation is influenced by the environment in which the plant is grown, as reported in canola [170] and winter wheat [171]. Indeed, from a breeding perspective, seed storage protein content is a quantitative trait which suggests that the phenotype is influenced by the environment. Research in soybean has demonstrated that differences in soil fertility were capable of increasing total seed protein content [172] as well as altering storage protein composition [135,173]. In canola, crop production on sulfur-limited land not only results in a reduction in glucosinolates and sulfur-containing amino acids [174,175,176] but can also alter seed storage protein profiles in the crop [177]. Recent work has also correlated nitrogen supply with the expression of seed storage protein genes in canola [114]. Though the extent to which environmental factors influence seed storage protein accumulation has yet to be empirically determined, the aforementioned studies suggest that agronomic practices can be exploited to alter seed protein content and quality.

## 11. Canola Protein as a Novel Food Product

Currently, canola protein is primarily used for animal feed but has the potential to be directly consumed in food products [64,178]. Canola protein does not have a history of human consumption, making it a novel food in the context of most food safety regulations. The procedures and standards for assessing the safety of novel foods, and the authorization required for their marketing, vary by country. Toxicology studies on the safety of a novel food are based on animal feeding experiments. While safety concerns were raised in early animal studies using rapeseed meal (non-canola quality), recent studies using canola-quality meal products concluded the ingredient to be safe. Collectively, the literature suggests that the safety of rapeseed meal was compromised by anti-nutritive compounds and the strict use of canola-quality *B. napus* alleviates safety concerns. Specific safety studies on a cruciferin-rich [179] and a napin-rich [180] canola protein isolate, respectively, found no adverse effects in rats fed with a diet supplemented with up to 20% of either product. Due to the high costs associated with acquiring regulatory approval, authorization of novel foods is typically only sought for countries with potential markets. Canola protein is currently a fledgling in the plant protein sector whose commercialization has been spearheaded by three companies; the history of canola protein commercialization and its challenges have been recently analyzed by Mupondwa et al. [64].

### 11.1. Allergenicity of Canola Seed Storage Proteins

The ability of napin to resist digestion suggests its potential as a food allergen [181,182]. Allergens are foreign proteins that are capable of evoking an adverse response from the immune system after they enter the body [183,184]. Generally, an immediate allergic response happens when an allergen has been encountered twice. The first time an allergen is encountered, the body produces a type of antibody that can bind the allergen called immunoglobulin E (IgE); this first encounter and production of IgE is collectively called sensitization [184]. Following sensitization, if the allergen is encountered a subsequent time, an immune response occurs in which symptoms typically associated with an allergic response, such as urticaria (hives) and rhinitis (runny nose), are experienced [183,184]. Repeated exposure to an allergen can result in increased severity of the symptoms. The molecular mechanisms behind how IgE function during allergies have been reviewed by Gould and Sutton [185].

Napin from canola has been identified as a potential allergen [128] based on its ability to bind IgE from sensitized patients [186]. Further, napin, in addition to cruciferin, have both been identified as allergenic proteins in cold-pressed canola oil that reacted with IgE from sensitized children [187]. These results were considered by the regulatory bodies when determining the safety of canola protein products, but were not deemed to be a substantial risk [188].

An allergenic reaction can be caused by a protein, to which a patient has not been previously sensitized, if it is sufficiently similar in structure to a known allergen [189,190]. This cross-reactivity suggests that patients who are allergic to other plant globulins and plant albumins, respectively, may react to cruciferin and napin and vice versa [182,190,191]. Mustard was added to a list of priority allergens in Canada [192] following a systematic review completed by Health Canada [193], who found sufficient scientific evidence of its allergenicity to be relevant to the Canadian public. The 2S albumin of mustard seed is the major allergen of mustard [191,194] and is similar in its amino acid sequence to napin [110,182], suggesting that people who are sensitized to mustard may also be allergic to napin.

Although the allergenicity of proteins in canola oil have been demonstrated and concerns of their potential to cross-react in patients sensitized to related protein allergens have been raised, it is noteworthy to consider that there have been no reports on allergies to canola oil, likely due to the refining process [187]. Furthermore, the long history of mustard oil consumption in certain parts of the world and the nutritional value of canola oil outweigh concerns of its allergenicity [195]; however, questions regarding the safety of concentrated canola protein products need to be evaluated, given their novel nature.

### 11.2. Regulation of Novel Foods in the United States

Canola meal and canola proteins have not had an extensive history of consumption and are considered novel foods by the United States Food and Drug Administration (FDA). Novel foods in the United States that are deemed generally recognized as safe (GRAS) by the FDA fall outside the purview of the Federal Food, Drug, and Cosmetic Act [196]. For a novel food to be granted GRAS status, its safety must be demonstrated empirically and be recognized as safe by the scientific community. Under the current GRAS Notification program, the FDA does not conduct scientific testing on the novel food to determine its safety, but rather, the Agency reviews the safety data it receives [196]. The general process required for an ingredient to acquire GRAS status involves three steps: first, the sponsor of the product performs or submits relevant scientific studies on the safety of the novel food in the form and dose it is intended to be used; second, the data are submitted as part of a GRAS notice to the FDA for review; and third, the FDA will respond with a letter stating whether or not it has questions regarding the GRAS conclusion of the novel food based on the submitted notice [196]. A letter indicating that the agency has no questions equates to acknowledgement and acceptance of the GRAS status of the ingredient. To date, six GRAS notices regarding *B. napus* products, of which three pertain to meal protein, and the FDA’s response to each one is found in the FDA’s online GRAS Notice Inventory (Table 6) (https://www.fda.gov/food/generally-recognized-safe-gras/gras-notice-inventory, accessed on 1 June 2021).

### 11.3. Regulation of Novel Food in the European Union

As in the United States, novel foods must be deemed safe before they can be used in the European Union. Canola protein and canola protein isolates are considered novel foods in the EU, which by definition are foods that are not “used for consumption to a significant degree” prior to 15 May 1997 (Regulation (EC) No 258/97) [197]. The safety determination process in the EU is generally similar to that of the GRAS Notification Program: sponsors of novel food submit data on the safety of the ingredient under its intended use to the European Commission, who then forwards the application to the European Food Safety Authority (EFSA). The EFSA assesses the data and publishes its finding as a Scientific Opinion; a committee votes whether to accept the finding; and the Commission issues authorization based on the vote. To date, the EFSA has published only one Scientific Opinion on a protein isolate derived from a mix of *B. napus* and *B. rapa*, finding the product to be safe under its intended use [188].

### 11.4. Regulation of Novel Food in Canada

In Canada, Health Canada is the regulatory body overseeing the safety of food additives and its decisions are enforced by the Canadian Food Inspection Agency (CFIA). Purified canola protein and canola protein isolates are considered novel foods in Canada as they do not have a history of use for human consumption [198]. The process to request approval for novel foods approximates that of the United States and European Union: first, the applicant submits pertinent scientific studies and data on the safety of the ingredient to the Food Directorate at Health Canada; second, the application is forwarded for scientific review after the Food Directorate has verified the information is complete; and third, the results of the review are sent back to the Food Directorate, who then decides on the authorization of the ingredient in question. Similar to the GRAS Notice Program, the decision of the Food Directorate is communicated as a Letter of No Objection for approved novel foods. An inventory of novel food decisions that received no objection (https://www.canada.ca/en/health-canada/services/food-nutrition/genetically-modified-foods-other-novel-foods/approved-products.html, accessed on 1 June 2021) is available online. Currently, no canola protein products have been approved as a novel food in Canada.

## 12. Canola Meal as a Protein Source in Animal Husbandry

Protein is an integral part of animal diets; however, it is typically a costly component of feed [199] and efforts to explore low-cost protein supplements have been untaken. The low cost and abundance of rapeseed and canola meal render it an economical protein source [33,64,200]. Mammals that are commercially raised for food production can largely be classified based on their ability to acquire nutrients from plant material: ruminants, such as cattle, have the capacity to ferment plant material prior to digestion allowing for successful nutrient uptake, while non-ruminants, such as swine, lack this ability [22]. This difference in digestive anatomy has implications in the use of canola meal as a protein supplement in the respective feeds of these animals: specifically, differing sensitivities to glucosinolates between ruminants and non-ruminants generally dictate the relative quantity of canola meal that can be incorporated into feed [51,201]. In addition to cattle and swine, poultry and fish are also raised for food on commercial scales and the feasibility of using canola meal in these production systems has also been studied.

### 12.1. Cattle

The United States is the largest importer of Canadian canola meal (Table 980-0012, Statistics Canada, 2019). The use of canola meal as a protein supplement has been widely adopted by the cattle (*Bos taurus* Linnaeus, 1758) industry since the development of canola [201]. Specifically, canola meal is one of the standard protein supplements for dairy cattle production [202]. Given the ruminant nature of cattle, the incorporation of plant-based protein supplements such as soybean meal is a well-established practice. A meta-analysis conducted on dairy cattle found that the use of canola meal was superior to soybean meal as a protein supplement as measured by feed intake, milk yield, and milk protein yield [203]. In cattle produced for meat, the substitution of barley as the protein source with canola meal during the growing period did not increase feed efficacy compared to the use of barley [204]. Although the conversion of feed to animal protein was not improved, the use of canola meal in feedlot cattle production may still be a more economical and sustainable alternative than using barley. Taken together, these studies support the continued use of canola meal as a protein supplement in cattle production.

### 12.2. Swine

Soybean meal is widely used as an economical protein source in swine (*Sus scrofa domesticus* Erxleben, 1777) production and remains a reference for novel plant-based protein supplements. Canola meal can partially replace soybean meal in the swine diet without ill-effect on animal health and production efficiency [75]; however complete replacement is hampered by its high fibre content and the presence of various secondary metabolites, especially glucosinolates based on feeding experiments [205]. Swine are more sensitive to glucosinolates compared to cattle [51,205] and residual glucosinolates in canola meal-containing feed will drive the animals to choose soybean meal-supplemented feed when presented with the option [206]. Additional processing steps during meal production such as toasting were able to mitigate feed rejection presumably due to the decomposition of residual glucosinolates; however, production efficacy did not supersede that of soybean meal [75]. Technical advances in canola meal processing will enable it to be a competitive and cost-efficient alternative to soybean meal [205]

### 12.3. Poultry

Poultry (*Gallus gallus domesticus* Linnaeus, 1758) ranks second as the most consumed animal protein globally [207]. Protein in the chicken diet is often supplied from a mix of animal and plant sources [199,208], the latter of which is primarily from soybean due to its nutritional properties [209]. Although chickens grow faster with some animal protein in their diet, work has been conducted to observe the effects of using solely vegetable-sourced protein for chicken [77,208,210,211,212]. Canola meal has been explored as a plant-based protein source for chicken production and was found to be generally comparable to soybean meal [212,213]. An examination of the lower digestive tract of chickens fed canola meal failed to identify intact proteins typical of canola seed, suggesting that the meal protein was completely digested by the animal [211]. Of cattle, swine, and chicken, the growth of the latter appears to be least affected by the use of canola meal as the sole source of feed protein, as evidenced by the possibility of using canola meal as the sole protein source in feed [83].

### 12.4. Aquaculture

Fish currently account for approximately 20% of the animal protein consumed globally and its production is split approximately in half between aquaculture (farmed fish) and capture fisheries (wild caught fish) [214]. The standard protein source for aquaculture has historically been fish meal; however, high costs, limited supply, and sustainability concerns led to a decline in the use of fish meal in favour of plant protein sources and other novel feedstuff [199,214,215]. The complete replacement of fish meal with other protein sources has been reported to be successful across various aquaculture species, particularly those of lower trophic levels [199]. As with land animals, fish are sensitive to glucosinolates and high levels in their feed from the inclusion of canola meal can negatively impact production efficiency [51]. Nonetheless, canola meal and different canola protein isolates have proven successful as a protein supplement in various aquaculture species [215], including rainbow trout (*Oncorhynchus mykiss* Walbaum. 1792) [216,217,218], and high-value crustaceans such as mitten crab (*Eriocheir sinensis* H. Milne-Edwards, 1853) [219] and shrimp species (*Litopenaeus stylirostris* Stimpson, 1871; *L. vannamei* Boone, 1931) [220,221].

## 13. Seed Storage Proteins in Food Processing

In addition to fulfilling its role as a macronutrient, protein can also be supplemented in food products to improve its nutritional profile. Plant-based protein isolates, namely derived from soybean and pea and comprised mostly of SSP, are ubiquitously available on the health food market in either its pure form or supplemented in food products [222]. The continued availability of these products on supermarket shelves speaks to the palatability of plant-based proteins and their acceptance by the general consumer.

Proteins also serve a non-nutritional role in food by directly controlling the physical and functional l food properties [64,198]. To ensure food products, particularly meat analogues, meet the textural expectations of consumers [223], commercial processors use different proteins [222] in addition to different mechanical processes [224]. As different SSP have different physiochemical properties based on their amino acid composition and consequently their structure, not all SSP can fulfill the same functional role [34,225]. More importantly, plant-based protein isolates contain a mixture of SSP whose constituents may be mutually antagonistic in their functionalities and raw isolates may therefore be unfit for use as a structural additive in commercial food processing despite its nutritive contribution [225]. One solution towards the adoption of plant protein for food processing applications is to supplement single SSP, or single SSP-enriched isolates rather than raw isolates. In this way, control over the texture of the final food product can be better maintained. Furthermore, SSP can be strategically used to improve not only the absolute protein content of the food product, but also the amino acid profile of the product [226,227], allowing for marketing to specific consumer bases.

## 14. Separation and Quantification of Seed Storage Proteins

Globulins and albumins can be separated based on their difference in solubility in dilute saline solution [115]. Indeed, napin and cruciferin have been successfully separated on the basis of solubility in salt [34,39,228,229,230]. Separation of cruciferin and napin can also be facilitated by the differences in solubility in different pH [73,96,228,229,230]. This appears to be the basis, at least partially, of patents granted for the commercial separation of these major canola SSP. A comprehensive review on separation technologies specific to canola protein and a selected list of relevant patents was recently compiled by Mupondwa et al. [64].

On a bench scale, the separation of cruciferin and napin can be accomplished by means of chromatography [98]. Although chromatographic separation and purification results in a highly purified single-SSP product, the cost of such methods, the technical expertise required for their operation, and the low recovery rate [96] limit such methods from being used on a commercial scale until broader market demand for these proteins develop. The ease of protein separation can be improved if the native SSP pool has reduced species complexity or consists of a single protein species.

Breeding efforts towards altering SSP composition require a method for quantifying individual canola proteins. Most breeding programs lack the expertise and infrastructure required to perform chromatography, and although accurate, such analytical methods lack the throughput required for phenotyping large populations. In current breeding programs, SSP quantification is performed by SDS-PAGE followed by Coomassie staining and densitometry [155]. However, this method fails to distinguish between different proteins with similar molecular masses and preferential binding of the dye to aromatic amino acids can skew the accuracy of the assay [231]. Immunological methods to quantify individual SSP may offer the accuracy and throughput needed for breeding programs.

## 15. Immunodetection and Quantification of Seed Storage Proteins

Electrophoretic separation of proteins followed by immunodetection of target antigens was first described by Towbin et al. [232] to analyze ribosomal proteins. To date, this variant of Western blotting remains a prevalent technique in the literature being employed in approximately 10% of all protein research articles [233]. Despite criticisms of its reproducibility due to variation in antibody quality [233], Western blotting can be a valuable technique for the relative quantification of proteins in mixtures provided proper extraction protocols, standard curves, and normalization protocols are implemented [234]. Specifically, the use of Western blotting to quantify individual SSP within total seed protein mirrors conventional size exclusion chromatography (SEC) methods [141] in that both techniques rely on the separation of proteins by molecular mass prior to estimating the abundance of target proteins of a given theoretical mass. Arguably, Western blotting requires less infrastructure to perform and the use of an antibody allows for greater discrimination of proteins compared to mass alone. Conversely, Western blotting has been implemented to qualitatively elucidate basic knowledge of SSP structure [127,235,236] and post-translational modifications [237]. Similarly, immunological methods have also been employed to study the spatial distribution of SSP [122] and quantify temporal patterns in their accumulation [98,238,239] through seed development.

## 16. Manipulation of Seed Storage Proteins in Select Crop Species

The challenges of chemical separation of SSP can be circumvented altogether with plants whose seed protein pool contains a single protein species. Seed storage proteins are regulated on a genetic level, and therefore genetic technologies can be employed to alter SSP composition in plants. Efforts were undertaken to genetically alter the SSP profile of various crop species, for functional [240,241] or nutritive purposes [240,242,243] have been successful.

### 16.1. Seed Storage Protein Manipulation through Conventional Breeding

Conventional breeding relies on cycles of crossing and selection to generate distinct genotypes with improvements in desired traits and genetic variability is required in plant breeding to facilitate genetic gain in the trait of interest [244]. To date, only one survey of SSP variability in rapeseed has been conducted in France [141], which found historic rapeseed cultivars to be relatively richer in napin content compared to modern canola-quality cultivars. Currently, the genetic variation that exists for SSP in Canadian germplasm remains unknown.

Diversity in SSP composition has been reported in various crop species such as wheat [245] and soybean [246], implying similar diversity may exist in canola. Wild soybean accessions with unique SSP profiles have been successfully incorporated into breeding programs to generate segregating populations with variable seed protein profiles [247] indicating that the manipulation of storage protein composition can be achieved by conventional breeding.

Intentional selection for SSP composition in canola has never been reported; however, selection for oil content is believed to have also inadvertently selected for increased cruciferin content [141]. A transcriptomic study of the breeding response in Chinese winter rapeseed over two decades of selection for improved fatty acid profiles and oil content revealed minimal changes to the transcription of cruciferin and napin, though actual levels of SSP were not determined [248]. Interestingly, protein and oil content both increased concomitantly with overall yield [248], suggesting that the inverse relationship between the two traits can be broken. Furthermore, cruciferin and napin levels were found to be highly heritable, suggesting the possibility for the successful genetic improvement of the levels of either protein through conventional breeding [155].

### 16.2. Soybean

Soybean is primarily grown as an oilseed in North America and as an important protein source in Asia [249]. Approximately 70% of the soybean SSP pool is composed collectively of the multimeric globulins glycinin (11S) and β-conglycinin (7S) [250,251]. While soy protein is versatile in both its functional and edible properties [252], a significant portion is directed towards tofu production. The marketability and consumer acceptance of tofu is mostly based on its texture; thus, research has focused on the relationship between soybean protein and its functional properties in tofu production. Seed storage protein composition is known to play a critical role in controlling the texture of tofu [253], and more recent work has demonstrated that the subunit composition of individual SSP also plays a critical role [254]. An effort to change the SSP profile of soybeans through a combination of mutagenesis and convention breeding has led to the development of genotypes with improved tofu-making qualities [247,254,255,256].

Simultaneously with improving protein functionality, efforts to improve the nutritional value of soybean protein have also been undertaken. The improvement of lysine content in the total seed protein of soybean was made possible by altering its feedback regulatory mechanism during biosynthesis [227]. Effort aimed at improving the methionine content of soybean protein was made through the expression of a chimeric gene encoding a methionine-rich δ-zein SSP from corn under the control of an endogenous soybean β-conglycinin promoter [257]. Increased accumulation of foreign proteins was enabled by the successful suppression of β-conglycinin production by RNAi [258,259]. These works demonstrate the possibility for the continued improvement of soy protein nutrition by enabling the improved accumulation of higher-nutrient or therapeutic proteins [259] at the expense of lower-nutrient endogenous SSP.

### 16.3. Wheat

In wheat, the seed protein pool, colloquially referred to as gluten, is comprised of the SSP gliadins and glutelins [109,260]. Wheat SSP primarily serves a structural rather than nutritive role in food and as such, breeding efforts have centered around improving its functionality. Despite the prevalence of wheat in the diet, the protein constituents of gluten elicits immunogenic reactions in susceptible patients and recent work has focused on the elimination of its reactivity [261,262]. Efforts to improve the functional properties of wheat have been made by various groups through the overexpression of glutenins [263,264,265]. The ectopic expression of genes encoding high-molecular-weight glutenin subunits resulted in an increased accumulation of the protein [264,265] as well as increased dough elasticity [263] in a dose-dependent manner. The downregulation of gliadins using RNAi has also been shown to improve the functionality of flour [266,267], demonstrating the direct relationship between SSP composition and protein functionality. Furthermore, flour from gliadin-deficient wheat genotypes was demonstrated to have reduced immunogenicity [261,262], suggesting that the manipulation of SSP can be a strategy to generate food products to cater to specialized dietary needs. Collectively, these studies in soybean and wheat provide empirical evidence on the possibility of manipulating seed storage proteins in canola.

### 16.4. Canola

To date, canola remains primarily an oilseed crop and its meal protein largely remains a byproduct [268]. Meal protein has only been considered for use as a nutritional supplement, and consequently efforts to alter the SSP profile of canola has only been examined from a nutritional standpoint. To improve the methionine content of canola meal protein, a chimeric gene containing the coding sequence of a methionine-rich 2S SSP from Brazil nut (*Bertholletia excels* Humb. and Bonpl.) under the control of either a phaseolin promoter [269] or soybean lectin promoter [270] (both promoters whose native functions drives SSP expression in the seed), was transformed into canola. Seeds from the resultant transgenic plants showed elevated levels of methionine [269,270], suggesting that the amino acid profile of canola meal protein was amenable to change and could be altered through the expression of foreign proteins in the seed. Efforts to improve the lysine content of canola meal protein were made by disrupting the feedback-regulation of lysine during biosynthesis, which effectively doubled the total seed lysine content [227]. This indicates that improvements to the levels of individual amino acids can be achieved. More importantly, these works suggest that the plant is able to accumulate higher than normal levels of individual amino acids without severe physiological consequences. Subsequent efforts to improve the quality of canola meal protein were made using advancements in antisense technology with the goal of selectively attenuating individual SSP [226,271]. By suppressing the accumulation of napin transcripts, an increase in cruciferin protein was observed without major effects on fatty acid composition [271]; similarly, reducing the level of cruciferin transcripts resulted in an increase in napin protein content as well as improved levels of methionine, lysine, and cysteine [226]. In both cases, modifications to the SSP did not result in changes to the total macromolecular profile of the seed. Interestingly, co-suppression of both cruciferin and napin together did not lead to a compensatory increase in oil [272], suggesting that the inverse relationship between oil and protein content is not a simple competition for metabolic intermediates.

## 17. Plant Breeding in the Omics Era

Advancements in omics technologies have enabled breeders to study the underlying mechanisms that govern desirable phenotypes on a genomic, transcriptomic, and proteomic level. These technologies improve the efficiency of breeding programs and improve the speed in which new cultivars can be generated.

### 17.1. Genomics

A major goal of genomics in crop agriculture is to correlate genotypic information with phenotypic data [273]. In this way, selections can be made early in the growth cycle on the basis of molecular markers that are impervious to environmental variation, improving both efficiency and accuracy. Early genotyping efforts relied on restriction digestion and hybridization to discern allelic variation. The advent of PCR subsequently enabled the development of amplification-based genotyping platforms with improved throughput and efficiency [273]. Next, genotyping by single-nucleotide polymorphism (SNP) markers was developed as the markers were numerous and widely distributed across plant genomes [274]. Multiplex SNP genotyping using crop-specific bead chip arrays such as the *Brassica napus* 60K Illumina Infinium™ SNP array [275,276] further enhanced the efficacy of acquiring genotypic information in crop plants [277] by enabling thousands of SNPs to be surveyed simultaneously. As next-generation sequencing becomes increasingly more accessible, genotyping with SNP markers may potentially be replaced by whole genome sequence data. While early whole-genome sequencing protocols such as restriction-site associated DNA sequencing [278] and genotyping-by-sequencing [279] used restriction enzymes to reduce the complexity of the genome and cost, rapid improvements in sequencing technologies now enable whole genomes to be economically sequenced at unprecedented speeds [280].

#### 17.1.1. Linkage Mapping

Linkage mapping identifies significant correlations between molecular markers and traits of interest in a structured population [281,282]; such populations are typically generated by crossing parents with divergent phenotypes and subsequently fixing recombination events in the segregating progeny through doubled-haploid production or repeated cycles of selfing. The size and relatedness between individuals in the population directly affect the resolution of the mapping study [283]. The mapping population along with the parental genotypes are phenotyped in replicated experiments and genotyped by SNP-chip arrays [284]. A linkage map displaying the physical order of the SNP markers and the genetic distance between them can then be built with various software packages by inferring recombination ratios [285,286]. Correlations between markers and phenotypic data are performed using various statistical methods [286]. Linkage mapping has been successfully employed in canola to identify molecular markers that co-segregate with total seed protein [145,146,147,148,149], seed storage protein content [155], and non-essential amino acid content [156].

#### 17.1.2. Association Mapping

Similar to linkage mapping, association mapping also aims to identify significant correlations between genotypic information and phenotypes [274,281]. However, unlike linkage mapping, which relies on recombination events occurring in a structure bi-parental population, association mapping (also referred to as genome-wide association studies; GWAS) takes into consideration all historic recombination events in a population of genetically diverse individuals [287]. The practical implications of this difference is twofold: first, by using a population of genetically diverse individuals, no time investment to generate biparental mapping populations is needed, which effectively shortens the time required for association mapping; second, by using genetically diverse genotypes and considering all historical recombination events, the resolution of association mapping is higher than that of linkage mapping, and causal SNPs can be identified rather than loci [281,287,288].

Genome-wide association studies were initially proposed for use in human genetics in the 1990s [289]. At a similar time, the first association studies were reported on grain crops [287]. Advancement in statistical methods to account for genetic challenges that are unique to crops were first incorporated into a GWAS for flowering time in maize a decade later [290]. To date, GWAS has been successfully and regularly implemented to study many major crop species [282,291,292].

The population size required for GWAS varies depending on the goal of the study and the inherent genetic structure of the population: 300 individuals are sufficient for candidate gene validation and 1000–5000 individuals are recommended for marker discovery [293]. Although larger populations are able to improve the power to detect QTL in canola by GWAS [294], populations of 200–500 individuals have been typical in recent studies (Table 7). Phenotypic data are collected from replicated experiments and genotypic data, often in the form of SNP markers, are typically acquired through high-throughput SNP-chip arrays that are crop specific [284]. Recent studies have used whole genome sequencing and transcriptome sequencing in place of SNP-chip arrays to generate genotypic data [295]. The significance of the association between each SNP and the phenotype is then tested using different statistical models while considering the relatedness between each individual of the population and possible cryptic population structures [274,282,287]. Finally, the results of GWAS are typically visualized in a Manhattan plot where the significance of the marker association is plotted against the physical position of the marker on the genome, thus offering a view of all the tested markers across the genome [296].

In canola, GWAS has been successfully implemented to identify molecular markers and loci associated with a variety of agronomically-important traits, and seed quality traits (Table 7); however, studies focusing on seed storage protein-related traits are lacking. In other major crops such as rice [297] and legumes [143], GWAS identified markers that underlie variation in storage protein accumulation. Furthermore, markers associated with amino acid content were identified using GWAS in maize [298], wheat [299], and soybean [300], as well as *Arabidopsis* [301]. The lack of similar association studies in canola focused on seed storage protein traits represents a void in the literature that warrants investigation.

## 18. Marking of a New Era of Crop Improvement with Genome Editing

Early efforts in the genetic manipulation of SSP profiles relied primarily on the use of antisense technology [226,240,271,316,317,318,319] and RNA interference [240,242,243,316,320,321]. These techniques relied on the degradation of gene transcripts to reduce the accumulation of select SSP. Advancements in genome editing technologies (succinctly reviewed by Langner et al. [322]), namely the Clustered regularly interspaced short palindromic repeats (CRISPR)/CRISPR-associated (Cas) system [323] and its derivatives, now enable the modification of SSP at the genetic level by inducing deleterious mutations in the associated genes. Briefly, the CRISPR/Cas9 system relies on a short single guide RNA (sgRNA) complex to guide the Cas9 endonuclease to the genomic region to which the former is complementary; after a double-stranded break is induced at the target site by Cas9, the host’s innate DNA repair mechanism repairs the break in an error-prone fashion, resulting in deleterious mutations that effectively silence the gene [323].

Initially applied to diploid animal and plant systems, CRISPR/Cas9 technology has also proven to be highly effective in targeting multiple homeologous alleles in polyploid *B. napus* [324]. Successful editing of the *ALCATRAZ* gene to improve shattering resistance [325], the *BnWRKY11* and *BnWRKY70* transcription factors to improve *Sclerotinia sclerotiorum* resistance [326], the *Fatty Acid Desaturase 2* gene to modify fatty acid profile [327] and other gene targets [328,329] in *B. napus* have recently been reported. Given the ease of use of the CRISPR/Cas9 system and its numerous successful applications in *B. napus*, genome editing technology has potential to be used as a tool to modify the SSP profile of canola.

Despite the increasing commercial interest in canola protein, breeding efforts towards improving canola meal protein quality are lacking. Research into the development of cultivars with altered SSP profiles will not only facilitate advancements in protein separation technologies, but also encourage the adoption and use of canola protein as a food product.

## Figures and Tables

**Figure 1 plants-10-02220-f001:**
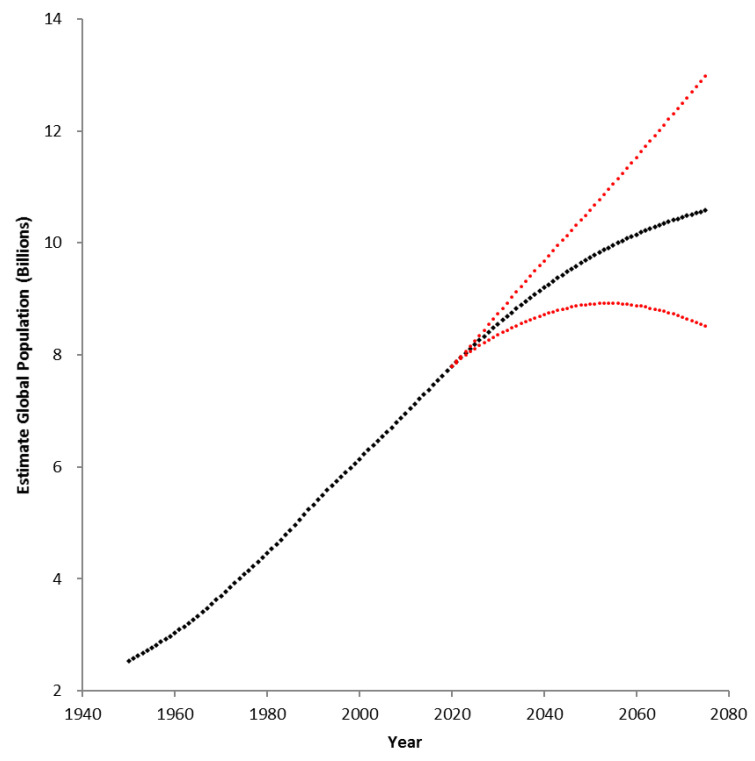
The world population is projected to reach nine billion by the mid-21st century. Data points up to 2020 are annual estimates. Black data points from 2021 to 2075 are estimates based on median fertility. Red data trends indicate high- and low-fertility variant estimates (raw data source: United Nations, Department of Economic and Social Affairs, Population Division (2019). World Population Prospects 2019, Online Edition. Rev. 1, accessed on 1 September 2021.

**Figure 2 plants-10-02220-f002:**
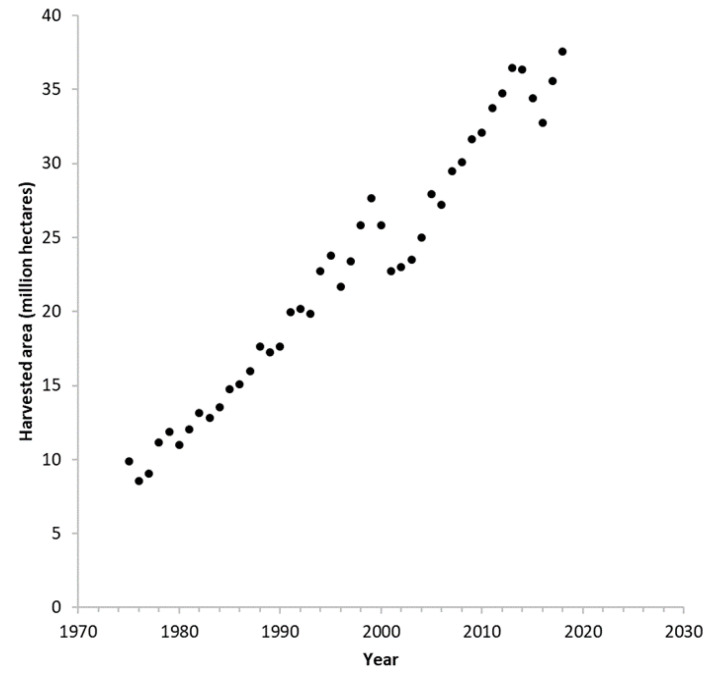
Global rapeseed (*Brassica napus* L.) harvested hectares have increased annually since the introduction of the first low-erucic acid, low-glucosinolate canola cultivar ‘Tower’ in 1974 (data source: FAOSTAT, Food and Agriculture Organization of the United Nations, accessed on 1 September 2021).

**Figure 3 plants-10-02220-f003:**
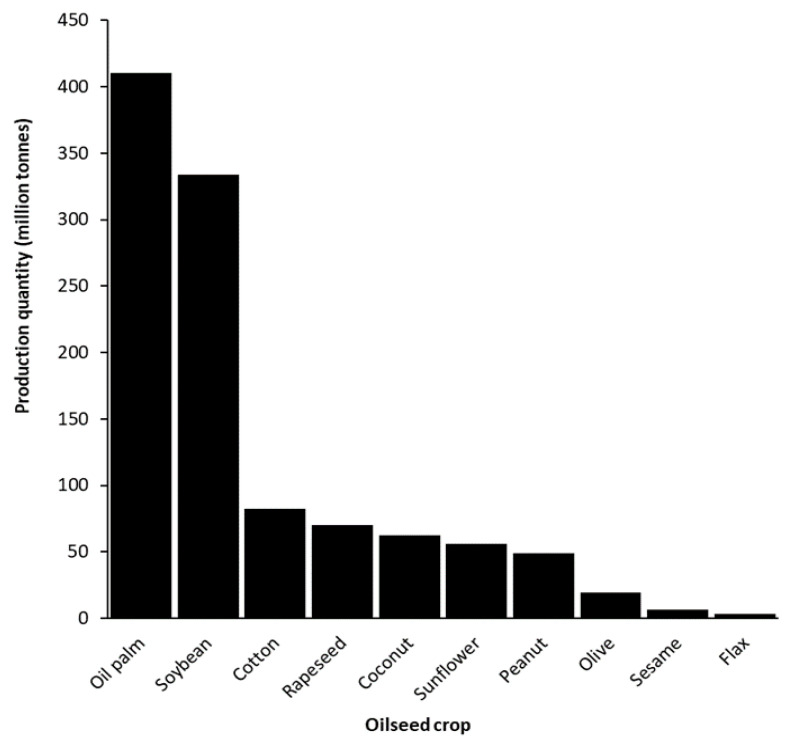
Rapeseed (including canola) was the fourth largest oil crop in 2019 based on global production quantity (data source: FAOSTAT, Food and Agriculture Organization of the United Nations, accessed January 2021).

**Figure 4 plants-10-02220-f004:**
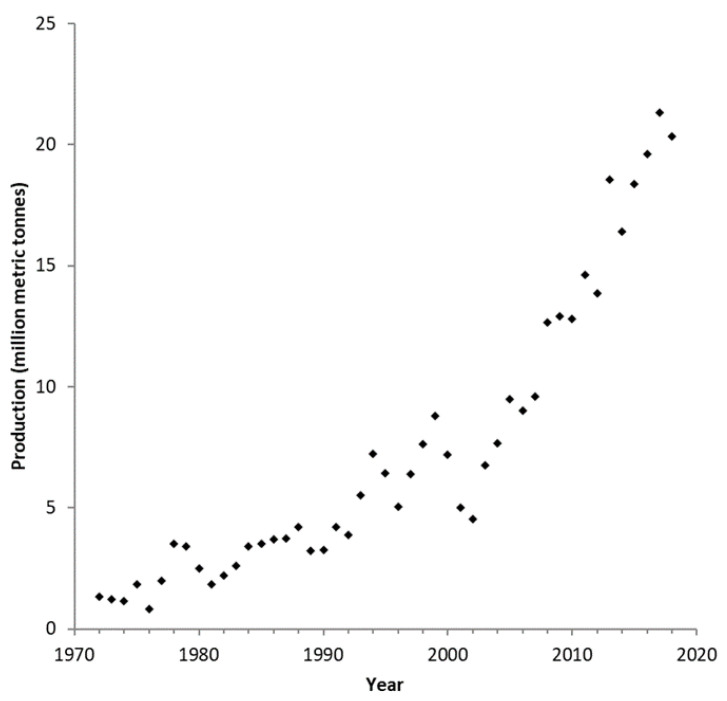
Canola production in Canada has increased steadily since the introduction of the first low-erucic acid, low-glucosinolate canola cultivar ‘Tower’ in 1974 (Source: Statistics Canada. Table 32-10-0359-01, accessed on 1 September 2021).

**Table 1 plants-10-02220-t001:** Comparison of seed quality between Canadian soybeans and western Canadian canola presented as the average value of 2013–2017 crops, inclusive. Table compiled with soybean data [82] and canola data [81] published by the Canadian Grain Commission. Non-applicable measurements indicated with n.a.

	Soybean	Canola
Protein content (%) ^1^	34.50	20.20
Oil content (%) ^1^	18.40	44.50
Oil-free protein of the meal (%) ^2^	43.80	37.80
Oleic acid (% in oil)	21.90	62.90
Linoleic acid (% in oil)	53.60	18.70
α-Linolenic acid (% in oil)	9.00	9.40
Total saturated fatty acids (% in oil)	15.20	6.70
Erucic acid (% in oil)	n.a.	0.01
Total seed glucosinolates (μmol/g, 8.5% moisture)	n.a.	10.00
Total glucosinolates of the meal (µmol/g, oil-free, dry basis)	n.a.	21.00

^1^ Based on 13% moisture in soybean and 8.5% moisture in canola. ^2^ Based on 13% moisture in soybean and 12% moisture in canola.

**Table 2 plants-10-02220-t002:** Comparison of moisture content, crude protein, and amino acid profiles between soybean meal and western Canadian canola meal. Values presented for canola meal are the mean and standard deviations of samples collected from five different crushing plants across western Canada. The origin and variability of the soybean meal were not disclosed [68].

	Soybean	Canola
Quality (%)		
Moisture	9.65	8.89 ± 0.43
Crude protein	46.10	37.82 ± 2.09
Amino acids (%)		
Alanine	1.97	1.64 ± 0.08
Arginine	3.27	2.15 ± 0.11
Aspartic acid	5.04	2.51 ± 0.12
Cysteine	0.61	0.81 ± 0.04
Glutamic acid	8.34	6.2 ± 0.41
Glycine	1.98	1.87 ± 0.10
Histidine	1.11	0.91 ± 0.05
Isoleucine	1.81	1.26 ± 0.09
Leucine	3.45	2.54 ± 0.14
Lysine	2.90	2.09 ± 0.08
Methionine	0.58	0.69 ± 0.03
Phenylalanine	2.25	1.44 ± 0.07
Proline	2.39	2.21 ± 0.11
Serine	2.31	1.47 ± 0.09
Threonine	1.73	1.52 ± 0.06
Tryptophan	0.69	0.52 ± 0.03
Tyrosine	1.74	1.06 ± 0.04
Valine	1.88	1.63 ± 0.09

**Table 3 plants-10-02220-t003:** Coincidence of zygotic embryogenesis stages in spring *Brassica napus* L. with spatial-temporal patterns in the accumulation of seed storage protein transcripts and gene products during the first six weeks seed of development under controlled growth environment conditions. Abbreviations are as follows: G, globular stage; H, heart stage; T, torpedo stage; C, cotyledon stage; CRU, cruciferin. A single asterisk (*) indicates initial detection and three asterisks (***) indicate peak.

Weeks after Pollination	1	2							3							4							5							6						
**Development**																																				
Crouch and Sussex (1981) Planta 153: 64–74									H				T					C																		
Custers et al. (1999) Protoplasma 208: 257–264		G		H			T		C																											
Fernandez et al. (1991) Development 111: 299–313												H							T					C												
Ilic-Grubor et al. (1998) Ann. Bot. 82: 157–165			H		T		C																													
Yeung et al. (1996) Int. J. Plant Sci. 157(1): 27–39					H		T									C																				
**CRU mRNA**																																				
DeLisle and Crouch (1989) Plant Physiol. 91: 617–623																		*														***				
Finkelstein et al. (1985) Plant Physiol. 78: 630–636															*																			***		
Sjodahl et al. (1993) Plant Mol. Biol. 23: 1165–1176																			*										***							
**CRU protein**																																				
Crouch and Sussex (1981) Planta 153: 64–74																			*					***												
**NAPIN mRNA**																																				
DeLisle and Crouch (1989) Plant Physiol. 91: 617–623																		*									***									
Finkelstein et al. (1985) Plant Physiol. 78: 630–636												*									***															
**NAPIN protein**																																				
Crouch and Sussex (1981) Planta 153: 64–74																			*					***												

**Table 4 plants-10-02220-t004:** Five manually curated entries for *Brassica napus* L. cruciferin are listed in the Uniprot database (Source: https://www.uniprot.org, accessed September 2021). Length is presented as amino acid count.

Entry	Entry Name	Protein Names	Length
P33522	CRU4_BRANA	Cruciferin CRU4 (11S globulin) (12S storage protein) [Cleaved into: Cruciferin CRU4 alpha chain; Cruciferin CRU4 beta chain]	465
P11090	CRUA_BRANA	Cruciferin (11S globulin) (12S storage protein) [Cleaved into: Cruciferin subunit alpha; Cruciferin subunit beta]	488
P33524	CRU2_BRANA	Cruciferin BnC2 (11S globulin) (12S storage protein) [Cleaved into: Cruciferin BnC2 subunit alpha; Cruciferin BnC2 subunit beta]	496
P33525	CRU3_BRANA	Cruciferin CRU1 (11S globulin) (12S storage protein) [Cleaved into: Cruciferin CRU1 alpha chain; Cruciferin CRU1 beta chain]	509
P33523	CRU1_BRANA	Cruciferin BnC1 (11S globulin) (12S storage protein) [Cleaved into: Cruciferin BnC1 subunit alpha; Cruciferin BnC1 subunit beta]	490

**Table 5 plants-10-02220-t005:** Seven manually annotated entries for *Brassica napus* L. napin are listed in the Uniprot database (Source: https://www.uniprot.org, accessed September 2021). Length is presented as amino acid length.

Entry	Entry Name	Protein Names	Length
P24565	2SSI_BRANA	Napin-1A (Napin BnIa) [Cleaved into: Napin-1A small chain; Napin-1A large chain]	110
P09893	2SSE_BRANA	Napin embryo-specific (1.7S seed storage protein) [Cleaved into: Napin embryo-specific small chain; Napin embryo-specific large chain]	186
P17333	2SS4_BRANA	Napin (1.7S seed storage protein) [Cleaved into: Napin small chain; Napin large chain]	180
P27740	2SSB_BRANA	Napin-B (1.7S seed storage protein) [Cleaved into: Napin-B small chain; Napin-B large chain]	178
P01090	2SS2_BRANA	Napin-2 (1.7S seed storage protein) [Cleaved into: Napin-2 small chain; Napin-2 large chain]	178
P01091	2SS1_BRANA	Napin-1 (1.7S seed storage protein) [Cleaved into: Napin-1 small chain; Napin-1 large chain] (Fragment)	133
P80208	2SS3_BRANA	Napin-3 (1.7S seed storage protein) (Napin BnIII) (Napin nIII) [Cleaved into: Napin-3 small chain; Napin-3 large chain]	125

**Table 6 plants-10-02220-t006:** Six canola (*Brassica napus* L.) products are listed in the United States Food and Drug Administration’s (FDA) Generally Recognized as Safe (GRAS) notice inventory since the inventory was established in 1998. In all applications, the FDA had no questions regarding the GRAS status of the product. Current as of September 2021.

GRAS Number	Notifier	Substance	Intended Use	Date of Closure	FDA Response
683	DSM Innovation Company	Canola protein isolate	Dietary protein Food processing	2017	no questions
682	Cargill Inc.	Lecithin from canola	Food processing Dietary fat Dietary choline	2017	no questions
533	American Lecithin Company	Lecithin from canola	Food processing Dietary fat Dietary choline	2015	no questions
425	Danone Trading B.V.	Canola oil (low-erucic-acid rapeseed oil)	Dietary fat (infant formula)	2012	no questions
386	BioExx Specialty Proteins, Ltd.	Canola protein isolate and hydrolyzed canola protein isolate	Food processing Dietary protein	2011	no questions
327	Archer Daniels Midland Company	Cruciferin-rich canola/rapeseed protein isolate and napin-rich canola/rapeseed protein isolate	Dietary protein	2010	no questions

**Table 7 plants-10-02220-t007:** Genome-wide association studies have been successfully implemented in *Brassica napus* L. to identify single-nucleotide polymorphism (SNP) markers associated with seed quality traits. Studies listed in this table were performed using populations of various sizes and genotypic data acquired using the Brassica 60K Illumina Infinium™ SNP genotyping array [276].

Trait	Population Size	Reference
Erucic acid content	215	[302]
Erucic acid content	203	[303]
Erucic acid content	472	[304]
Fatty acid composition	435	[305]
Fatty acid composition	520	[59]
Fatty acid content	370	[306]
Fibre content	520	[307]
Glucosinolate	521	[308]
Glucosinolate	203	[303]
Glucosinolate	520	[309]
Glucosinolate	1425	[310]
Glucosinolate	203	[311]
Glucosinolate	203	[312]
Glucosinolates	215	[302]
Oil content	472	[304]
Oil content	521	[313]
Oil content	105	[314]
Oil content	370	[59]
Oil content	521	[313]
Oil content	203	[311]
Oil content	203	[315]
Protein content	370	[59]

## Data Availability

All data in this review has been retrieved from previous publications and cited.

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
