# Peer review of "Breeding Canola (Brassica napus L.) for Protein in Feed and Food"

_plants, 2021, doi:10.3390/plants10102220_

Round 1
Reviewer 1 Report
A Review on the Use of Canola Protein for Feed and Food
- The title of this manuscript refers to use of canola meal for feed and food. I had expected to read about applications of meal in food but it was very much weighted to discussion about genetic manipulation. In the 20 sections, below, there is a relatively small proportion given to canola for food. Nutritional value and antinutrients consist of one page, protein in animal husbandry a bit over 1 page, and very little on food use. Separation and measurement of protein is restricted to one page. Two sections seemed to be missing (13 and 14) in my version of the text. Numerous references listed at the end of the paper are also missing from the text.
Approx. heading Page No.
- Intro 1
- Rape to canola 3
- Extracting oil 5
- Effect of processing 5
- Nutritional Value 6
- Antinutrients 7
- Seed development 8
- Storage proteins 9
- Genetic control of ssp 11
- Non-genetic control 12
- Canola protein as a novel food 12
- Canola meal in animal husbandry 15
- ?
- ?
- Ssp in processing 16
- Separation and quantification 17
- Immunodetection 17
- Manipulation of ssp in other crops 18
- Omics 19
- Genome editing 22
- Generally, the paper is well written and well referenced. Section 5 however contains duplicated text which has been copied and pasted twice into the same paragraph. There is discussion about Table 1 but I couldn’t find reference to it in the text. The comparison of two crops, soybean and canola, at different moisture content and from different growing environments is disappointing considering the influence of environment on production and, particularly, on oil and protein.
- I have made some individual comments below concerning particular corrections I think need to be made within the text. There may be instances where I have misunderstood the context and I am sure there are others that I have missed.
- It is difficult to say if the number of references is necessary considering this is a review. However, if references are listed in the Reference section but not used in the text, they should be removed. Additionally, considering the paper relates to food use, I question the large number of references related to genetics compared to those for food use, protein extraction, processing and measurement.
- Overall, I would consider the paper should be published with prior editing. If the paper proceeds with the current content I would recommend that the title be changed to indicate the level of discussion about breeding and genetic manipulation and less about food use.
Pg. 3, line 104: I think this should say “.. before rapeseed meal could be incorporated…”.
Pg. 5, line 161: canola and rapeseed meal can be differentiated based on glucosinolate content. Purified protein agreeably will not be able to be differentiated.
Section 5
Pg 6 line 191-200: There appears to be no reference to Table 1.
Pg 6, line 199: Superscript “1” needs formatting
Pg 6 line 201: The line needs formatting.
Pg 6, line 204: The line needs formatting
Page 6, line 208-220: This section needs to be deleted.
Pg 6, line 200 (and Table 1): Comparisons of soy and canola protein have been done at different moisture levels. Although these percentages are used in industry to measure crop quality, it is done for a purpose. Since you are comparing two products in a table, why wouldn’t you normalise these to the same moisture content to make a sensible comparison?
Pg 7, Table 1, last line: Change to “Total glucosinolates of the meal (µmol/g, oil-free, dry basis)”
Pg 7 line 230: The origin of the soybean meal is important when describing a comparison between two crops, highly influenced by environment. It would be useful to know, for example, if the soybean was at least grown in western Canada or somewhere entirely different.
Pg 8, line 251-294: I question the relevance of this detail on seed development in a review of use of protein for food. Perhaps it could be reduced to a reference on seed development?
Pg 9, line 296: Seed storage protein (SSP)…
References
Pg 31, line 1368: “Meeting…..”
Pg 31, line 1401: “Realigning…”
The last 4 pages of the manuscript contains over 100 references and it is questionable as to the necessity of all of these, considering this paper is about the “Use of Canola Protein in Food” and this section is about plant breeding.
In particular, there appear to be about 20 references in the reference section which don’t appear to have been used in the text. i.e. 352, 353, 354, 355, 356, 357, 358, 359, 360, 361, 374, 375, 376, 377, 378, 379, 380, 381, 382, 383.
The references should be reduced to reflect those relevant to the topic and all those not required need to be removed.

Reviewer 2 Report
I found this review easy to read and informative – a worthwhile addition to the scientific literature. Some requested corrections:
Figure 1: Data were accessed almost 2 years ago. Please update.
Figure 2: Data were accessed almost 2 years ago. Please update. In legend, replace ‘acres’ with ‘hectares’. Consider replacing with global production rather than cultivation area.
L92: Correct species name to ‘oleracea’
L94: Unhelpful focus on Western Canada. If it is unhealthy in Canada it is unhealthy everywhere.
L108: B. napus is a cultigen with no known ‘wild’ rapeseed. Maybe replace with ‘industrial’ or ‘original’ rapeseed.
Figure 4: Again, the focus on Canada is unhelpful for an international readership.
L187-220: This whole section is duplicated.
The footnotes from Table 1 seem to be inserted too.
L204: ‘higher fiber’ compared to what?
L208: PDCAAS has already been explained in L194.
Table 1: 2 decimal places appears to be unwarranted in terms of precision since the last digits are all ‘0’ except for erucic acid.
Table 2: In the legend, also comment on the lack of variability information available for soybean. Alternatively, seek out a better source of information, it must be commonplace for such a major protein crop.
L231 onwards: This paragraph should include a short acknowledgement of the differential sensitivities of monogastric and ruminant animals to these compounds. This is taken up in a later section but I think it is helpful to flag here in this section.
Figure 5: Plant development is highly responsive to temperature. Either convert to thermal time (in degree-days) or explain the temperature ranges used by the cited studies.
L334/Table 3: Two years ago. Please re-check and update. Italicise species name in table header.
Table 4: Same.
L367: If copy number is important, Southern blotting is obsolete given the availability of several reference genomes. Please provide this information or remove.
L435-436: This is a problematic and unnecessary sentence and should be removed.
L574: Be consistent in how species authorities are cited. Using dates is not conventional except in some taxonomic contexts. Why not simply ‘Bos taurus L.’?
L587: Same
L602: Same And other species – please address all of these.
L798-885: Detailed review of genetic analysis methodology is beyond the scope of this review and should be summed up in one short paragraph.
Table 6 should either be removed or at most include seed composition traits only
Author Response
I found this review easy to read and informative – a worthwhile addition to the scientific literature. Some requested corrections:
Figure 1: Data were accessed almost 2 years ago. Please update.
Updated
Figure 2: Data were accessed almost 2 years ago. Please update. In legend, replace ‘acres’ with ‘hectares’. Consider replacing with global production rather than cultivation area.
Updated and corrected to hectares
L92: Correct species name to ‘oleracea’
Corrected
L94: Unhelpful focus on Western Canada. If it is unhealthy in Canada it is unhealthy everywhere.
Western has been removed.
L108: B. napus is a cultigen with no known ‘wild’ rapeseed. Maybe replace with ‘industrial’ or ‘original’ rapeseed.
Replaced ‘wild’ with ‘industrial’
Figure 4: Again, the focus on Canada is unhelpful for an international readership.
Since canola was initially developed in Canada, I felt it was fitting to track the production within the country to present day.
L187-220: This whole section is duplicated.
The footnotes from Table 1 seem to be inserted too.
Duplication and footnotes have been removed
L204: ‘higher fiber’ compared to what?
Fixed incomplete comparison
L208: PDCAAS has already been explained in L194.
Fixed
Table 1: 2 decimal places appears to be unwarranted in terms of precision since the last digits are all ‘0’ except for erucic acid.
I agree the extra precision is unnecessary. The terminal zeros were added to make the values align in decimal places to the erucic acid value
Table 2: In the legend, also comment on the lack of variability information available for soybean. Alternatively, seek out a better source of information, it must be commonplace for such a major protein crop.
Lack of variability included in legend
L231 onwards: This paragraph should include a short acknowledgement of the differential sensitivities of monogastric and ruminant animals to these compounds. This is taken up in a later section but I think it is helpful to flag here in this section.
Differential sensitivities acknowledged
Figure 5: Plant development is highly responsive to temperature. Either convert to thermal time (in degree-days) or explain the temperature ranges used by the cited studies.
All the studies were conducted in growth chambers using temperatures (20-25 oC days) typical for canola studies. This has been noted in the legend.
L334/Table 3: Two years ago. Please re-check and update. Italicise species name in table header.
Changed
Table 4: Same.
Changed
L367: If copy number is important, Southern blotting is obsolete given the availability of several reference genomes. Please provide this information or remove.
Change to “initially estimated” and included comment “no updated estimates have been published despite the availability of reference genomes.”
L435-436: This is a problematic and unnecessary sentence and should be removed.
L574: Be consistent in how species authorities are cited. Using dates is not conventional except in some taxonomic contexts. Why not simply ‘Bos taurus L.’?
Zoological references usually include dates in the author description, but botanical references do not. I followed the zoological nomenclature examples (https://www.iczn.org/the-code/the-code-online/ section 51) for the animals names.
L587: Same
As above
L602: Same And other species – please address all of these.
As above
L798-885: Detailed review of genetic analysis methodology is beyond the scope of this review and should be summed up in one short paragraph.
The title for the article has been amended to “Breeding Canola (Brassica napus L.) for Protein in Feed and Food” to expand the scope of the article slightly to encompass the extra information.
Table 6 should either be removed or at most include seed composition traits only
Table 6 has not been removed as the authors feel is it still an important part of the review.
Reviewer 3 Report
This review is about the use of canola (Brassica napus L.) protein for feed and food. The authors discussed the current uses of canola meal in animal husbandry are presented to underscore potential limitations for the consumption of canola meal in mammals. General discussions on the allergenic potential of canola proteins and the regulation of novel food products are provided to highlight some of the challenges that will be encountered on the road to commercialization and general acceptance of canola protein as a dietary protein source.
The manuscript is very well written, and I found no major faults in the manuscript. Congratulation to the author for writing such a good review. It can be accepted after some minor changes.
L104 incorporated in the human diet to incorporated into the human diet.
L300 storage-stable proteins which to storage-stable proteins, which.
L308 work of Osborne [116] who to work of Osborne [116], who.
L327 hexameric protein, whose subunits to hexameric protein whose subunits.
L329 into octameric structure to into the octameric structure.
L339 peptide is detached resulting to peptide is detached, resulting.
L341 glycosylation occur to glycosylation occurs.
L386 soluble thus enabling to soluble, thus enabling.
L400 winter rapeseed [141] while similar studies to winter rapeseed [141], while similar studies.
L477 can result in increasing to can result in increased.
L491 Health Canada [195] who found to Health Canada [195], who found.
L491 napin [70,184] suggesting to napin [70,184], suggesting.
L502 need to be evaluated given their novel nature to need to be evaluated, given their novel nature.
L534 assesses the data and publish to assesses the data and publishes.
L548 the Food Directorate who to the Food Directorate, who.
L610 canola seed suggesting to canola seed, suggesting.
L627 species [217] including to species [217], including.
L641 different mechanical process to different mechanical processes.
L702 Efforts undertaken to genetically alter the SSP profile to Efforts were undertaken to genetically alter the SSP profile.
L709 conducted in France [141] which found to conducted in France [141], which found.
L725 with overall yield [253] suggesting to with overall yield [253], suggesting.
L727 heritable suggesting to heritable, suggesting.
L767 immunogenicity [266,267] suggesting to immunogenicity [266,267], suggesting.
L783 meal protein were made by to meal protein was made by.
L796 increase in oil [277] suggesting to increase in oil [277], suggesting.
L840 mapping which relies to mapping, which relies.
L842, L847 all historic recombination to all historical recombination.
L892 and it derivatives, now enable to and its derivatives now enable.
Author Response
This review is about the use of canola (Brassica napus L.) protein for feed and food. The authors discussed the current uses of canola meal in animal husbandry are presented to underscore potential limitations for the consumption of canola meal in mammals. General discussions on the allergenic potential of canola proteins and the regulation of novel food products are provided to highlight some of the challenges that will be encountered on the road to commercialization and general acceptance of canola protein as a dietary protein source.
The manuscript is very well written, and I found no major faults in the manuscript. Congratulation to the author for writing such a good review. It can be accepted after some minor changes.
L104 incorporated in the human diet to incorporated into the human diet.
Changed
L300 storage-stable proteins which to storage-stable proteins, which.
Changed
L308 work of Osborne [116] who to work of Osborne [116], who.
Changed
L327 hexameric protein, whose subunits to hexameric protein whose subunits.
Changed
L329 into octameric structure to into the octameric structure.
Changed
L339 peptide is detached resulting to peptide is detached, resulting.
Changed
L341 glycosylation occur to glycosylation occurs.
Changed
L386 soluble thus enabling to soluble, thus enabling.
Changed
L400 winter rapeseed [141] while similar studies to winter rapeseed [141], while similar studies.
Changed
L477 can result in increasing to can result in increased.
Changed
L491 Health Canada [195] who found to Health Canada [195], who found.
Changed
L491 napin [70,184] suggesting to napin [70,184], suggesting.
Changed
L502 need to be evaluated given their novel nature to need to be evaluated, given their novel nature.
Changed
L534 assesses the data and publish to assesses the data and publishes.
Changed
L548 the Food Directorate who to the Food Directorate, who.
Changed
L610 canola seed suggesting to canola seed, suggesting.
Changed
L627 species [217] including to species [217], including.
Changed
L641 different mechanical process to different mechanical processes.
Changed
L702 Efforts undertaken to genetically alter the SSP profile to Efforts were undertaken to genetically alter the SSP profile.
Changed
L709 conducted in France [141] which found to conducted in France [141], which found.
Changed
L725 with overall yield [253] suggesting to with overall yield [253], suggesting.
Changed
L727 heritable suggesting to heritable, suggesting.
Changed
L767 immunogenicity [266,267] suggesting to immunogenicity [266,267], suggesting.
Changed
L783 meal protein were made by to meal protein was made by.
Changed
L796 increase in oil [277] suggesting to increase in oil [277], suggesting.
Changed
L840 mapping which relies to mapping, which relies.
Changed
L842, L847 all historic recombination to all historical recombination.
Changed
L892 and it derivatives, now enable to and its derivatives now enable.
Changed